Genome-Enhanced Detection and Identification (GEDI) of plant pathogens

Feau Nicolas 1
Beauseigle Stéphanie 2
Bergeron Marie-Josée 3
Bilodeau Guillaume J. 4
Birol Inanc 5
Cervantes-Arango Sandra 1
Dhillon Braham 6
Dale Angela L. 1 7
Herath Padmini 1
Jones Steven J.M. 5 8 9
Lamarche Josyanne 3
Ojeda Dario I. 10
Sakalidis Monique L. 11
Taylor Greg 5
Tsui Clement K.M. 12
Uzunovic Adnan 7
Yueh Hesther 1
Tanguay Philippe 3
Hamelin Richard C. richard.hamelin@ubc.ca 1 13
1 Department of Forest and Conservation Sciences, Forest Sciences Centre, University of British Columbia , Vancouver , BC , Canada
2 Biopterre , Sainte-Anne-de-la-Pocatière , Quebec , Canada
3 Canadian Forest Service, Natural Resources Canada , Quebec city , Quebec , Canada
4 Ottawa Plant Laboratory, Canadian Food Inspection Agency , Ottawa , Ontario , Canada
5 BC Cancer agency, Genome Sciences Centre , Vancouver , BC , Canada
6 Department of Plant Pathology, University of Arkansas at Fayetteville , Fayetteville , AR , United States of America
7 FPInnovations , Vancouver , BC , Canada
8 Department of Medical Genetics, University of British Columbia , Vancouver , BC , Canada
9 Department of Molecular Biology and Biochemistry, Simon Fraser University , Burnaby , BC , Canada
10 Department of Biology Unit of Ecology and Genetics, University of Oulu , Oulu , Finland
11 Department of Plant, Soil & Microbial Sciences and Department of Forestry, Michigan State University , East Lansing , MI , United States of America
12 Faculty of Medicine, University of British Columbia , Vancouver , BC , Canada
13 Foresterie et géomatique, Institut de Biologie Intégrative des Systèmes, Laval University , Quebec city , Quebec , Canada
Solomon Peter
Electronic publication date: 2018 Feb 22
Publication date: 2018
Volume: 6
Electronic Location ID: e4392
Received 2017 Nov 7; Accepted 2018 Jan 29
Copyright year: 2018
License: This is an open access article, free of all copyright, made available under the Creative Commons Public Domain Dedication. This work may be freely reproduced, distributed, transmitted, modified, built upon, or otherwise used by anyone for any lawful purpose.
License URL: https://creativecommons.org/publicdomain/zero/1.0/

Keywords: Plant pathogens, Genomics, Detection and identification, Diagnostics, Mycology, Plant pathogen

Funding: Genome Canada Genome British Columbia Canadian Forest Service Genomics Research and Development Initiative, GRDI Canadian Food Inspection Agency Large Scale Applied Research Project (LSARP) 2112 This work was funded by Genome Canada, Genome British Columbia, the Canadian Forest Service (Genomics Research and Development Initiative, GRDI), Canadian Food Inspection Agency, FP Innovations and Boreal Genomics through a Large Scale Applied Research Project (LSARP 2112; Genome Canada) grant. The funders had no role in study design, data collection and analysis, decision to publish, or preparation of the manuscript.

==============================
Plant diseases caused by fungi and Oomycetes represent worldwide threats to crops and forest ecosystems. Effective prevention and appropriate management of emerging diseases rely on rapid detection and identification of the causal pathogens. The increase in genomic resources makes it possible to generate novel genome-enhanced DNA detection assays that can exploit whole genomes to discover candidate genes for pathogen detection. A pipeline was developed to identify genome regions that discriminate taxa or groups of taxa and can be converted into PCR assays. The modular pipeline is comprised of four components: (1) selection and genome sequencing of phylogenetically related taxa, (2) identification of clusters of orthologous genes, (3) elimination of false positives by filtering, and (4) assay design. This pipeline was applied to some of the most important plant pathogens across three broad taxonomic groups: Phytophthoras (Stramenopiles, Oomycota), Dothideomycetes (Fungi, Ascomycota) and Pucciniales (Fungi, Basidiomycota). Comparison of 73 fungal and Oomycete genomes led the discovery of 5,939 gene clusters that were unique to the targeted taxa and an additional 535 that were common at higher taxonomic levels. Approximately 28% of the 299 tested were converted into qPCR assays that met our set of specificity criteria. This work demonstrates that a genome-wide approach can efficiently identify multiple taxon-specific genome regions that can be converted into highly specific PCR assays. The possibility to easily obtain multiple alternative regions to design highly specific qPCR assays should be of great help in tackling challenging cases for which higher taxon-resolution is needed.

Introduction

Plant diseases caused by fungi and Oomycetes represent worldwide threats to crops and forest ecosystems. The inadequacy of disease diagnostics, reporting protocols and the lack of centralized recording mechanisms have been identified as causes for the increase in emerging diseases of plants and animals (Fisher et al., 2012). To prevent and manage those emerging diseases, rapid detection and identification of the causal pathogens are crucial. This is particularly important for tree diseases where large contiguous forest ecosystems can be threatened by invasive and emerging pathogens, resulting in ecosystem-wide irreversible damage when prevention and management fail (Pautasso, Schlegel & Holdenrieder, 2015). Plant pathogens can be transmitted by a variety of means, including natural dispersal in water, rain and wind. But because of their cryptic nature and their ability to remain dormant in plants or soil, introduction and spread of alien invasive species are often related to anthropogenic activities. The intensification of drivers such as international trade, combined with climate change contribute to the emergence and rapid increase of the threat that plant pathogens may cause to ecosystems (Desprez-Loustau et al., 2016; Pautasso, Schlegel & Holdenrieder, 2015; Santini et al., 2013).

DNA detection methods have provided powerful tools with broad applications for the detection and monitoring of invasive species. However, DNA-based assay design for plant pathogen detection relies mostly on the amplification and detection of a very small fraction of the target organism’s genome. Assays are generally designed to amplify between one and three genes or genome regions that are conserved in all eukaryotes such as the internal transcribed spacer region (ITS rDNA), intergenic spacer region (IGS) and beta-tubulin (BT) (Schena et al., 2013). Single nucleotide polymorphisms (SNPs) in these conserved regions are sometimes abundant enough to design primers and/or probes that are specific to a targeted taxon and can discriminate closely related species (Prévost-Bouré et al., 2011; Thonar, Erb & Jansa, 2012; Schena et al., 2013). However, this method has limitations when applied to taxa that diverged recently since they often share highly homologous or even identical sequences in these conserved gene regions. This limits assay design and increases the risk of false positive results by making it difficult to find discriminant SNPs. This could be critical in cases where closely related microorganisms pose different risks and lead to different epidemiological or regulatory outcomes.

Genomics can play a valuable role in the discovery and characterization of emerging pathogens (Firth & Lipkin, 2013). The increase in genomic resources brought by next generation sequencing opens the way to mining entire genomes of pathogens and their close relatives to identify genes or genomic regions that are unique to a taxon or to a group of taxa. This is promising and has been used to design assays for a wide range of microbe-host combinations. Applications include detecting genetic material shed by invasive fish in environmental (eDNA) water samples (Farrington et al., 2015), human infecting bacteria (Ho et al., 2011; Ho et al., 2012; Hung et al., 2012), zoonotic infectious bacteria (Hänsel et al., 2015; Blaecher et al., 2017) and plant pathogenic bacteria (Pritchard et al., 2016).

Herein, we describe a pipeline to compare whole genomes of plant pathogens and phylogenetically related taxa to identify genome regions specific to targeted taxa or groups of related taxa and design highly accurate PCR assays. As a proof-of-concept, we have applied our assay development pipeline to a range of taxonomically diverse plant pathogens to demonstrate that it is robust, flexible and broadly applicable. Our method allowed for the rapid and efficient discovery of genome regions that provide highly specific targets for the development of assays to detect some of the most damaging plant pathogens across three broad taxonomic groups: Phytophthora (Heterokonts, Oomycota), Dothideomycetes (Fungi, Ascomycota) and Pucciniales (Fungi, Basidiomycota). This pipeline can be applied to a broad range of organisms, including pathogens of crops, forest trees and animals for which genomic resources are available.

Materials and Methods

Pipeline development

Our approach required access to assembled and annotated genomes of the targeted organisms as well as closely related taxa. The underlying principle of our method is to compare the protein content within the genomes of phylogenetically related taxa to ensure the selection of targets that are discriminant towards the most closely related known species. Our bioinformatics pipeline is divided into four modules (Fig. 1). In the first module, genomic resources (i.e., sets of gene and protein models predicted from a whole genome sequence) are selected for the targeted species under investigation and for a group of related species. In the second module, clusters of homologous genes are generated from the protein sets of all available genomes to identify candidate protein clusters that are unique to the targeted species. The third module is a filtering step aimed at eliminating false positive candidates generated in the previous step. The fourth module consists of automated primer and probe design for the candidate genes retained in module 3. The four modules are detailed below and the pipeline is applied to plant pathogens.

Figure 1 Pipeline for development of qPCR assays using whole genomes.

Module 1: genomic resources

In order to maximize the accurate identification of candidate protein clusters that are unique to the targeted taxa and contain enough polymorphisms for the design of species-specific or group-specific qPCR assays, genomes of target and related non-target taxa are required (e.g., same genus and order). To achieve this, de novo genome assemblies and full protein sets are either produced and assembled for the targeted species under investigation and for a group of related species or recovered from public genome data repositories. Next-generation sequencing (NGS) techniques constitute a fast and cost-effective way of obtaining whole genome sequences, particularly in eukaryotic organisms with small genomes such as pathogenic fungi (Faino & Thomma, 2014). De novo genome assemblies can quickly be achieved with assemblers such as ABySS (Simpson et al., 2012; Gallo et al., 2014; Abbas, Malluhi & Balakrishnan, 2014) (see : ‘Computational testing and experimental screening’); then, AUGUSTUS (Stanke et al., 2006) can provide a fast way to generate ab initio gene/protein model prediction (i.e., the “gene space”) from these de novo assemblies (Ma & Fedorova, 2010; Haas et al., 2011; Faino & Thomma, 2014). In addition, a collection of genomes of target and non-target taxa can be obtained through the publicly available fungal genome sequencing initiatives such as the Mycocosm (Grigoriev et al., 2014) and the 1000 Fungal Genomes project (http://1000.fungalgenomes.org/home/). Currently, more than 2,100 de novo fungal genome assemblies are available in the NCBI database, representing about one thousand fungal species distributed along all the branches of the fungal tree of life (Fig. 2).

Figure 2 Number and phylogenic coverage of fungal (A) and Oomycete (B) genomes available on the NCBI public database.

Module 2: discovering homologous gene clusters

OrthoMCL (Li, Stoeckert & Roos, 2003) is used to generate clusters of homologous genes (orthologs and paralogs) using the protein sets of all genomes previously selected. This allows for identification of candidate protein clusters that are unique to a taxon or that are common to members of a group and absent in the other taxa (“higher hierarchical levels”). Briefly, the OrthoMCL procedure comprises an all-against-all BLASTp step with a similarity cutoff defined by an e-value of 1e–20 and minimum overlap of 50% followed by a Markov clustering (MCL) of groups of homologs. Cluster tightness is regulated by an inflation parameter arbitrarily selected between values of 1.1 (fewer clusters with more proteins in each) to 5.0 (smaller clusters of proteins with high similarity); by default, we used I = 1.5 to try to cluster together as many homologous proteins with relatively high similarity. Following MCL clustering the output file is parsed to identify candidate clusters. These can be taxa-specific clusters containing one or several proteins unique to a single taxon or higher level hierarchical clusters containing at least one protein homolog for each taxon in a group (e.g., multiples species within a clade); cDNA sequences are then retrieved from each candidate protein cluster, aligned using MUSCLE (Edgar, 2004) and stored in a fasta format until further processing.

Module 3: filtering false positives

This filtering step is crucial to account for putative unique genes that are present in other genomes but were mis-annotated and/or fragmented. Gene fragments may be caused by truncation of the protein-coding sequence in the genome due to the termination of a contig in the middle of an open reading frame. Such fragments may result in OrthoMCL clustering errors by clustering them in groups that are distinct from the group containing their true full-length homologs. Thus, to ensure that candidate clusters were truly unique, i.e., specific to the target genome, BLASTn and BLASTp searches were conducted using candidate proteins and cDNA sequences retrieved in Module 2 against all non-target genomes and protein sets, respectively (Fig. 1). The BLASTn search minimizes the likelihood of obtaining a candidate cluster that would result from an annotation error in the target genome or due to gene mispredictions in the non-target genomes. The BLASTp search aims to minimize the effect of truncated protein models (Text S1).

The e-value cutoff for BLASTp and BLASTn searches must be selected based on the level of identity between the genomes of target and non-target taxa, the estimated number of candidates, the quality of the genomes and the tolerance of potential false positives. Filtering simulations (see Results section ‘Computational testing’) helped to identify default parameters with “stringent” cutoff threshold values set at 1e–20 for both BLASTn and BLASTp to retain a minimum of candidate clusters.

Module 4: assay design

The final module implements the assay design, using PRIMER3 (Rozen & Skaletsky, 1999) to generate unique or common primers and probes for PCR or qPCR assays. For each candidate gene, a primer pair and a probe are automatically designed from the cDNA alignment with PRIMER3 parameters as described in Table 1.

Table 1 Oligonucleotide primer and probes default parameters used in qPCR automated assay design (Module 4).

	Optimal	Range	Other	
Amplicon size	–	80–150 bp	–	
Primers				
Length	20 nt	18–24 nt	–	
Melting temperature (Tm)	60 °C	60–63 °C	–	
GC%	–	50–60%	–	
Maximum Tm difference between primers	1.0	–	–	
Maximum 3′ self-complementary	1.0	–	–	
Maximum poly X	3.0	–	–	
3′-end		–	No G or C	
Hybridization probe				
Length	–	18–30 nt	–	
Melting temperature (Tm)	–	65–70 °C	–	
GC%	–	45–65%	–	
Maximum poly X	3.0	–	–	
5′-end	–	–	No G or C	

Computational testing and experimental screening

Testing the cutoff filters

False positive filtering with BLASTn and BLASTp (Module 3) was first tested with a set of 11 Phytophthora genomes and subsequently using a second set of 16 Dothideomycetes genomes. For each dataset, Module 2 was run with default parameters. Then, filtering simulations with different BLASTn and BLASTp cutoff values were carried out as described in Text S1 (“Filtering simulations with Module 3”).

Pipeline runs

We assessed the robustness of this pipeline by testing it in three divergent taxonomic groups. The genomes of 19 tree pathogens belonging to the genus Phytophthora, the class Dothideomycetes or the order Uredinales were sequenced, assembled and annotated (Tables  S1–S3). Paired-end Illumina sequencing, read filtering and assembly procedures were described in Lamarche et al. (2015) and Feau et al. (2016). Ab initio gene and protein model predictions were obtained by using AUGUSTUS trained with (i) de novo transcriptome assemblies obtained using trans-ABySS for the corresponding species of Phytophthora and Dothideomycetes (Text S1 “RNA-seq libraries”) or (ii) gene and protein sets from closely related species for the Pucciniales. From each of these three genome collections, a subset was selected, which included the targeted species and other species in the same taxonomic group. This subset was then enriched with publicly available genomes from additional phylogenetically related species.

We tested the pipeline on the invasive species Phytophthora ramorum responsible for the sudden oak death and sudden larch death diseases, P. lateralis a pathogen of Port Orford Cedar (Chamaecyparis lawsoniana) and P. kernoviae a pathogen of european beech (Hansen et al., 2000; Brasier et al., 2005; Brasier & Webber, 2010). In order to perform genome comparison to identify unique genome regions, we obtained the genomes of eight non-target taxa from phylogenetic clades 1, 2, 7, 8 and 10 (as defined in Martin, Blair & Coffey, 2014) (Table S3). In the Dothideomycetes, we obtained the genomes of 39 Dothideomycetes (mostly in the Capnodiales and one Pleosporales; Table S1) and searched for candidate genes in two poplar pathogens causing cankers (Sphaerulina musiva) and leaf spots (S. populicola; Dhillon et al., 2015) and in Phaeocryptopus gaeumannii, the causal agent of the Swiss needle cast of Douglas fir.

For the rust fungi, we obtained the genome sequences of 17 Pucciniales, including members of the genera Melampsora, Cronartium, Endocronartium, Puccinia, Mixia, Sporobolomyces and Rhodotorula (Table S2). We searched these genomes for candidate genes in three rusts attacking poplars and pines: the European (Melampsora larici-populina) and North American (M. medusae f. sp. deltoidae) poplar rust and the white pine blister rust (Cronartium ribicola).

In silico screening for intra-taxon variation

We assessed the possibility that some of the candidate genes obtained in Module 3 may fail to generate the expected amplicon in all the individuals within the targeted taxa due to intra-taxa variability and presence/absence polymorphisms. BLASTn searches (e-value cutoff of 1e–20) were conducted using the candidate genes obtained for P.  ramorum after Module 3 filtering against 40 P. ramorum de novo genome assemblies (AL Dale, N Feau, SE Everhart, B Dhillon, B Wong, J Sheppard, GJ Bilodeau, A Brar, JF Tabima, D Shen, CM Brasier, BM Tyler, NJ Grunwald & RC Hamelin, 2013, unpublished data). Similarly, candidate genes targeting S. musiva were searched against 16 de novo genome assemblies obtained for this species (ML Sakalidis, N Feau, B Dhillon & RC Hamelin, 2013, unpublished data). We then conducted some simulations to identify the number of candidate genes that would need to be multiplexed to successfully target all individuals within the target species. This consisted in searching the de novo assemblies (BLASTn e-value cutoff of 1e–20) with combination patterns of one to ten candidate genes. For each pattern, 500 random combinations were generated and tested.

Experimental screening

For each group of plant pathogens, we selected a subset of the candidate genes identified by the pipeline. Candidate genes from different scaffolds were selected to maximize the representation of genomic regions. Candidate genes were screened with biological material to eliminate genes that failed to generate the expected amplicon in the targeted species or generated amplicons in the most closely related species. Screening was conducted by performing standard PCR using the selected primers and DNA of the target and the most closely related species, followed by gel electrophoresis.

DNA was extracted from seventeen Dothideomycetes cultures representing 14 Capnodiales species, including five Sphaerulina species closely related to S. musiva and S. populicola (Feau, Hamelin & Bernier, 2006). For testing the candidate genes obtained for P. ramorum and P. lateralis, we used DNA extracted from 41 species distributed among nine phylogenetic clades (out of the ten clades known; Martin, Blair & Coffey, 2014), including 11 species from clade 8 which are closely related to the two targeted species. This collection was reduced to 22 species for testing P. kernoviae-specific candidate genes, but included the other two species that are found in clade 10 (P. boehmeriae and P. gallica) with P. kernoviae. DNA from rust samples was obtained from 19 Melampsora and 12 Cronartium and Endocronartium taxa and 18 species from three additional Uredinales genera. For the Phytophthora and Dothideomycetes, 5 ng of DNA template was used in the PCR reactions (95 °C for 3 min, 29 cycles of 95 °C for 30 s, 62 °C for 30 s and 72 °C for 1 min, and a final extension step of 72 °C for 10 min) in a 25 µl volume reaction mix (10 µM primer concentration and 0.2 µl Platinum Taq DNA polymerase; Invitrogen, Life Science Technologies, Carlsbad, CA, USA). For the Pucciniales, primer concentration was at 1 µM and thermocycling conditions were slightly different : 94 °C for 3 min, 35 cycles of 92 °C for 30 s, 60 °C for 30 s and 72 °C for 1 min, and a final extension step of 72 °C for 10 min. PCR amplification was assessed running 5 µl of the PCR product in a 1.5–2% agarose gel stained with ethidium bromide.

Results

Computational testing

Filtering simulations

Simulations with BLASTn and BLASTp filtering conducted on a set of 13 Phytophthora and 16 Dothideomycetes genomes suggested that the BLASTp e-value cutoff had a positive impact in removing false positive candidates when the genome sequences used were of poor quality (i.e., “fragmented genomes”; Text S1 “Filtering simulations with Module 3”, Figs. S1 and S2). When genomic distances between target and non-target taxa are short, rejection of candidate by the BLASTn filter can rapidly increase with diminution of the e-value cutoff (Fig. S3B). Use of stringent filters (i.e., high e-value cutoffs) rejected most of the potential false positive candidates, in particular when the genomic distances between the target and non-target taxa were short i.e., when taxa were closely related (Text S1 “Filtering simulations with Module 3”, Fig. S3).

Pipeline run on Phytophthora

Nine Phytophthora genomes were selected in Module 1 (Table 2; Feau et al., 2016). OrthoMCL search in Module 2 identified 52,280 clusters, including 1,624 (3.1%) and 5,578 (10.7%) putative unique clusters in P. ramorum and P. lateralis, respectively (Table  2). The filtering step in Module 3 decreased the number of candidates to 37 for P. ramorum and 180 for P. lateralis, suggesting that about 97% of the putative unique candidate clusters predicted in the OrthoMCL analysis were false positives according to our filtering criteria (Table 2). At a higher hierarchical level, the OrthoMCL search resulted in 110 clusters that were shared between P. ramorum and P. lateralis and not found in the six other species; nine (0.02%) of these clusters were retained after the Module 3 filtering with default values. For P. kernoviae, 830 putative unique clusters were identified out of 44,422 OrthoMCL clusters, which were reduced to 55 candidates after Module 3 filtering.

Table 2 Number of species and genus-specific protein models found after clustering with OrthoMCL and filtering with BLASTn.

Targets	Module 1	Module 2	Module 3	
	Non-target genomes	# of protein models	# OrthoMCL clusters	# OrthoMCL unique clusters	# of unique clusters	
Phytophthora						
Species-specific						
P. ramorum	PINFa, PSOJ, PLAT, PCAP, PCIN, PHIB, PFOL		52,280b	1,624 (3.1%)	37	
P. lateralis	PINF, PSOJ, PRAM, PCAP, PCIN, PHIB, PFOL		52,280b	5,578 (10.7%)	180	
P. kernoviae	PINF, PSOJ, PRAM, PCAP, PCIN		44,422	830 (1.9%)	55	
Group-specific						
P. ramorum+P. lateralis	PINF, PSOJ, PCAP, PCIN, PHIB, PFOL		52,280	110 (0.2%)	9	
Dothideomycetes						
Species-specific						
Sphaerulina musiva	STON, SPOP, MPUN, PGAE, MGRAM, MLAR, MGIB, DPIN, DOTS, MDEAR, CLAF, MFIJ	160,617	44,189b	885 (2.0%)	131	
S. populicola	STON, SMUS, MPUN, PGAE, MGRAM, MLAR, MGIB, DPIN, DOTS, MDEAR, CLAF, MFIJ	160,617	44,189b	765 (1.7%)	134	
Phaeocryptopus gaeumannii	STON, SMUS, SPOP, MPUN, MGRAM, MLAR, MGIB, DPIN, DOTS, MDEAR, CLAF, MFIJ, CERZ, BAUCO, DIDZE	193,449	51,248b	3,350 (6.5%)	1,000	
Group-specific						
S. musiva+S. populicola	STON, MPUN, PGAE, MGRAM, MLAR, MGIB, DPIN, DOTS, MDEAR, CLAF, MFIJ	160,617	44,189	392 (0.9%)	68	
S. musiva+S. populicola+ STON1	STON, MPUN, PGAE, MGRAM, MLAR, MGIB, DPIN, DOTS, MDEAR, CLAF, MFIJ	160,617	44,189	345 (0.8%)	163	
Rusts						
Species-specific						
Melampsora larici-populina	All Pucciniales genomes in Table S3, but M. larici-populina		13,355b	3,550 (26.6%)	1,519	
M. medusae f. sp. deltoidae	All Pucciniales genomes in Table S3, but M. medusae f. sp. deltoidae		20,713b	8,901 (43.9%)	1,542	
Cronartium ribicola	All Pucciniales genomes in Table S3, but C. ribicola		9,633b	2,782 (28.8%)	1,341	
Genus-specific						
Melampsora genus	All Pucciniales genomes in Table S3, but Melampsora spp.		1,870c	374 (20.0%)	270	
Cronartium genus	All Pucciniales genomes in Table S3, but Cronartium spp.		1,027c	51 (5.0%)	34	
Notes.

a Species name abbreviations: PINF, Phytophthora infestans; PSOJ, P. sojae; PCAP, P. capsicii; PCIN, P. cinnamomi var. cinnamomi; PHIB, P. hibernalis; PFOL, P. foliorum; PRAM, P. ramorum; PLAT, P. lateralis; STON, Mycosphaerella sp. STON; SMUS, Sphaerulina musiva; SPOP, S. populicola; PGAE, Phaeocryptopus gaeumannii; MPUN, Ramularia endophylla, MGRAM, Zymoseptoria tritici; MLAR, M. laricina; MGIB, Pseudocercospora pini-densiflorae; DPIN, Dothistroma pini; DOTS, D. septosporum; MDEAR, Lecanostica acicula; CLAF, Cladosporum fulvum; MFIJ, M. fijiensis; CERZ, C. zeae-maydis; BAUCO, Baudoinia compniacensis; DIDZE, Didymella zeae-maydi.

b Number of OrthoMCL clusters that include at least one gene from the targeted species.

c Number of OrthoMCL clusters that include at least one gene for each species of the genus.

Pipeline run on Dothideomycetes

Genomic resources for 13 Dothideomycete fungal species were selected in Module 1 of the pipeline (Table 2). Statistics for the de novo genome assemblies were within the range of those obtained from public databases with N50 ranging from 0.06 to 0.18Mb and BUSCO completeness over 99.3% (Table S1). From 160,617 protein models, OrthoMCL generated 16,103 protein clusters and 28,086 singletons, with 885 and 765 unique clusters for S. musiva and S. populicola, respectively. This represents about 8.0% of the protein content predicted for these two genomes (Dhillon et al., 2015). Module 3 filtering reduced this number to 131 and 134 candidates for S. musiva and S. populicola, respectively. At higher hierarchical levels, analysis of poplar pathogens S. musiva, S. populicola and Mycosphaerella sp. STON1 resulted in 163 candidate clusters (52.8% of the candidate clusters were eliminated) whereas for the two pathogens of the poplar section Aigeiros, S. musiva and S. populicola, 82.6% of the 392 candidate clusters were discarded. Adding three Dothideomycete genomes to our initial collection before searching candidates for the Swiss needle cast fungus P. gaeumannii increased the total number of clusters to 16,662 (17% increase) and singletons (34,586) found by OrthoMCL (Table 2). Among these, 3350 (6.5%) were unique in silico candidates for P. gaeumannii and 1,000 candidate clusters remained after Module 3 filtering.

Pipeline run on Pucciniales

In Module 1, a comprehensive proteome dataset from 17 rust species (including nine obtained in this project) was selected to identify candidates unique to the individual species within two rust fungi genera, Melampsora (M. larici-populina and M. medusae) and Cronartium (C. ribicola), and genes that are conserved at the genus level for these two groups. OrthoMCL clustering (Module 2) resulted in a similar proportion of candidate clusters in M. larici-populina (3,550 clusters; 26.6% of all clusters) and C. ribicola (2,782; 28.8%) (Table 2). The number of unique clusters obtained for M. larici-populina was only 9% lower than what was obtained in a pairwise comparison with the wheat rust Puccinia graminis f. sp. tritici (3,903 unique gene clusters) (Duplessis et al., 2011), suggesting that addition of new proteomes to the clustering analysis does not have a drastic impact on the number of species-specific clusters. A high proportion of unique clusters (about twice the number found in the other two rust species) was obtained for M. medusae f. sp. deltoidae, likely resulting from the high number of protein models predicted in the genome of this pathogen (Table 2). After filtering in Module 3 the number of unique genes was reduced to 1,519, 1,542 and 1,341 for M. larici-populina, M. medusae f. sp. deltoidae and C. ribicola, respectively (Table 2). Filtering at the genus level resulted in only 270 candidate clusters shared between seven Melampsora species and 34 between the four Cronartium/Endocronartium species (Table 2 and Table S3). Module 4 rejected 73.3% to 76.4% (for Melampsora and Cronartium, respectively) of these candidates due to the presence of interspecific polymorphisms preventing the design of primers and probes with PRIMER3.

In silico screening for intra-taxa variation

Twenty-one out of the 37 candidate genes (56.7%) obtained for P. ramorum were successfully retrieved in the 40 de novo assemblies tested for P. ramorum (Fig. 3A). The observed level of presence/absence of the candidate genes may be explained by polymorphism among the divergent P. ramorum lineages. Simulations carried out on the de novo assemblies with random combinations of one to nine of the 40 candidate genes indicated that a minimum of four candidate genes have to be multiplexed to successfully target the expected amplicon in 99.99% of the P. ramorum individuals (Fig. 3B). For S. musiva, 82% of the candidate genes were found in all 16 de novo genome assemblies tested. For this taxon, simulations indicated that the combination of three random candidate genes should be sufficient to successfully target all S. musiva individuals (Figs. 3C and 3D).

Figure 3 In silico screening for intra-taxon variation in Phytophthora ramorum (A and B) and Sphaerulina musiva (C and D).

Number of candidate genes predicted for P. ramorum (n = 37) and S. musiva (n = 134) that targeted different proportions of the de novo genome assemblies of P. ramorum (n = 40) (A) and S. musiva (n = 16) (C). Minimum number of candidate gene required to successfully target all the de novo assemblies of P. ramorum (B) and S. musiva (C).

Experimental screening

For each targeted taxon or group studied, primer pairs for 5 to 65 candidate genes identified in the pipeline were evaluated for specificity. Overall, 76 (25.5%) candidate genes were validated out of the 297 tested. Within each group, the proportion of retained candidate genes after wet lab testing for specificity were in the same range (22.6, 28.0% and 33.3% for the Pucciniales, Dothideomycetes and Phytophthora, respectively) and not significantly different from the overall proportion of successful candidates (Chi-2 value = 1.76, P [3df] = 0.62) (Table 3) (Figs. S4 and S5). Furthermore, the number of non-target species considered in each group for PCR-testing (11–24 species for the Pucciniales, 11–14 species for the Dothideomycetes and 22–40 species for Phytophthora; Table 3) did not impact this success rate, as suggested by the lack of correlation between these two variables (R2 =  − 0.06, P = 0.55). Similarly, the quality of the de novo genome assembly (genome N50-value; Tables  S1–S3) used for the targeted taxa does not seem to impact this success (R2 =  − 0.06, P = 0.61).

Table 3 Experimental screening of the candidate clusters unique to species or group of taxa.

Targeted taxa	# tested targeted taxa	# tested non-targeted taxa	# candidate genes tested	# success	
Phytophthora					
P. ramorum	11 P. ramorum	40 Phytophthora spp.	28	5 (17.9%)	
P. lateralis	4 P. lateralis	40 Phytophthora spp.	16	6 (37.5%)	
P. kernoviae	1 P. kernoviae	22 Phytophthora spp.	12	9 (75.0%)	
P. ramorum+P. lateralis	11 P. ramorum, 4 P. lateralis	39 Phytophthora spp.	19	5 (26.3%)	
Dothideomycetes					
Sphaerulina musiva	2 S. musiva	14 Mycosphaerella spp.	51	14 (27.5%)	
S. populicola	2 S. populicola	14 Mycosphaerella spp.	65	16 (24.6%)	
Phaeocryptopus gaeumannii	10 P. gaeumannii	14 Mycosphaerella spp.	10	3 (30%)	
S. musiva+S. populicola	2. S. musiva, 2 S. populicola	12 Mycosphaerella spp.	39	13 (33.3%)	
S. musiva+S. populicola+Mycosphaerella sp. STON1	2. S. musiva, 2 S. populicola, 1 Mycosphaerella sp. STON1	11 Mycosphaerella spp.	6	2 (33.3%)	
Rusts					
Melampsora larici-populina	13 M. larici-populina	15 Melampsora spp., 1 Coleosporium sp., 1 Pucciniastrum sp., 1 Cronartium sp., 2 Chrysomyxa spp.	10	2 (20%)	
M. medusae f. sp. deltoidae	10 M. medusae	15 Melampsora spp., 1 Coleosporium sp., 1 Pucciniastrum sp., 1 Cronartium sp., 2 Chrysomyxa spp.	10	2 (20%)	
Cronartium ribicola	10 C. ribicola	10 Cronartium spp., 5 Melampsora spp., 3 Coleosporium spp., 3 Pucciniastrum spp., 2 Chrysomyxa spp.	20	3 (15%)	
Melampsora genus	19 Melampsora spp.	2 Coleosporium spp., 3 Pucciniastrum spp., 3 Cronartium spp., 3 Chrysomyxa spp.	5	3 (60%)	
Cronartium genus	11 Cronartium spp.	8 Melampsora spp., 4 Coleosporium spp., 5 Pucciniastrum spp., 7 Chrysomyxa spp.	8	2 (25%)	

Discussion

We provide a proof-of-concept for using whole genome sequence comparisons to design detection and diagnostic assays at different hierarchical levels in taxonomically diverse groups of crop and tree pathogens. The 73 whole genome sequences generated in this project or obtained from public databases were parsed to discover genes that are either unique to individual taxa or shared at different hierarchical levels within groups of taxa. Our approach, comprising in silico design and in vitro validation steps, generated 85 assays designed to specifically target species or groups of related species belonging to the Phytophthoras, Dothideomycetes and Pucciniales.

Having access to whole genome sequences provides an unprecedented opportunity to explore levels of variability among genomes, genes and taxa. With the expansion of the comparative genomics field and the development of tools such as Markov clustering (Li, Stoeckert & Roos, 2003; Szilagyi & Szilagyi, 2014), pairwise BLASTp and BLASTn (Lin et al., 2010; Yang et al., 2013; Xu et al., 2015), identification of the “non-core genome”, i.e., taxon-specific regions and genes, has become routine in computational genomics. Gene content-based differentiation between closely related taxa of filamentous microorganisms with different lifestyles and ecology was crucial to identify genomic factors underlying these traits and gain insights into the mechanism of genome evolution (Dhillon et al., 2015; Goodwin et al., 2011; Grandaubert et al., 2014; Plissonneau, Stürchler & Croll, 2016). We used a similar conceptual approach to identify taxon-specific genome regions and validated the hypothesis that these regions could serve as targets for the development of molecular assays for taxa detection.

Most PCR assays for pathogen detection use genes that are conserved to allow for primer design but comprise polymorphisms which can be targeted for design of discriminant oligonucleotide probes or primers. However, there is a trade-off in the selection of genome regions between low intra-specific heterogeneity (to allow design of universal primers) and interspecific divergence (to allow taxa discrimination). The internal transcribed spacer region (ITS) of the nuclear ribosomal repeat unit is widely used for PCR assays as well as for taxa discrimination and in DNA barcoding and metabarcoding studies (Schoch et al., 2012). It has the advantages of providing a large database of sequences and occurring in multiple copies in genomes which enable the development of very sensitive detection assays. However, the ITS can be limited for discrimination of phylogenetically related species and specificity is often an issue (Gazis, Rehner & Chaverri, 2011). In particular, ITS resolution for taxa separation in species-rich Ascomycota (e.g., genera Cladosporium, Penicillium and Fusarium) has often been inferior to several of the protein-coding genes commonly used in fungal taxonomy (Seifert, 2009; Feau et al., 2011; Schoch et al., 2012; Vialle et al., 2009). Similarly, some of the taxa within the groups that were used in this study are closely related and not readily distinguishable in assay design using the ITS region. Only eight SNPs [98.0% similarity] are found between the ITS sequences of S. musiva and S. populicola and six SNPs [99.03% similarity] are present between the ITS sequences of P. ramorum and P. lateralis (Feau et al., 2005; Werres et al., 2001). In addition, sub-specific lineages with different epidemiological and biological characteristics often have nearly identical ITS sequences, complicating or preventing the design of reliable and robust taxon-specific assays. This is the case of the four lineages of P. ramorum that differ in mating types and aggressiveness yet share identical ITS sequences (Ivors et al., 2004; Eyre et al., 2014).

The method described herein provides benefits compared with assays conventionally designed on the basis of SNPs found in conserved genes among taxa. By focusing on taxon-specific genes for qPCR detection assays, we expected to lower type I errors, thereby maximizing specificity and reducing false positives. Despite its relative simplicity, our pipeline comprises several steps to limit both false positives and negatives. Filtering conducted in Module 3 greatly reduced the number of candidate genes by performing reciprocal blast searches across the genomes. This step was essential because of the variation in quality encountered in genome assembly and annotation. A second in vitro screening eliminated candidate genes that failed to amplify or discriminate close relatives. This two-step approach was efficient in quickly identifying potential unique and shared genes that are candidates for assay design and eliminating those that could yield false positives. It should be noted that the rate of conversion from in silico to in vitro validation steps (∼28%) could be improved by fine-tuning the assay design and conducting additional in silico testing for intra-taxa variability using additional de novo assemblies in the targeted taxa. Moving or redesigning primers would make it possible to obtain amplicons for those candidate genes that failed the in silico validation step. However, since we obtained a large number of candidate genes for each taxonomic group targeted and our goal was to establish a proof-of-concept, we simply eliminated those that failed the in vitro screening step and we were still able to generate a large number of assays.

Another advantage of our approach is that primer design is less constrained since the genes targeted are unique, or at least they are not found in the genomes of phylogenetically related taxa. This could facilitate the design of multiplex qPCR assays that use internal probes with different fluorochromes within a single reaction. Development of multiplex assays requires homogenization of annealing temperatures for the primers and probes targeting each amplicon. This design should be more efficient since the primers and probes in the unique targeted genes can be moved or their size can be customized without having the constraint of targeting discriminant SNPs. In addition, targeting unique genes reduces the likelihood of cross-interaction amongst the amplicons.

The conceptual and practical simplicity of qPCR, in combination with its speed and sensitivity, have made it the technology of choice in many diagnostic applications, including microbial quantification (Yu et al., 2005; Narihiro & Sekiguchi, 2011; Thonar, Erb & Jansa, 2012) and pathogen detection (Orlofsky et al., 2015). Multiple qPCR assays have become extremely powerful to detect fungi, bacteria and parasites (Kamau et al., 2014; Gosiewski et al., 2014; Bilodeau et al., 2009) and we believe that this pipeline should enable development of qPCR assays in crop and tree pathogens. The pipeline described herein could be easily applied to any DNA-based detection methods (Yeo & Wong, 2002), such as loop-mediated isothermal amplification (LAMP), hybridization-based microarray (Huang et al., 2006) or PCR-ELISA (Loeffler et al., 1998). It would be particularly well-suited to target-enrichment methods by providing a large number of relevant genomic regions that can be used to enrich pathogen target genes in environmental samples. We expect that the impact of our approach and its efficiency in developing taxon-specific targets will improve quickly with the steady increase in the number of whole fungal and oomycete genome sequences (Fig. 2).

Given the wealth of genomic resources available and the increase in next-generation sequencing of microorganisms, this approach promises to be very useful to overcome some of the limitations of DNA detection assays based on conserved orthologous genes. Our work is a clear demonstration that a genome-wide approach can be useful to efficiently identify multiple, taxon-specific gene regions at different hierarchical levels that contain reliable priming site. We successfully discovered a large number of unique genes even between closely related species such as P. ramorum and P. lateralis (99.0% similarity in the ITS) and S. musiva and S. populicola (98.0% similarity). This approach is therefore likely to be broadly applicable to other fungi and Oomycetes. The possibility to easily obtain multiple alternative regions with equal or improved performance to the ITS region and other universal protein-coding genes commonly used for designing qPCR assays should be of great help in dealing with challenging cases for which higher taxonomic resolution is needed, such as for hybrids and races. Heteroploid organisms such as Phytophthora x alni (Aguayo et al., 2016) represent more complicated cases in the development of qPCR diagnostic tools (e.g., Martin et al., 2012). Homology to the putative parental species (Ioos et al., 2005; Inderbitzin et al., 2013) and deficiency of concerted evolution in homogenizing intra-individual copies (Lindner et al., 2013) in the ITS region make this marker often unsuitable for assay design in such organisms. Multiple taxon-specific assays derived from whole genome sequences that target DNA inherited from the different lineages should help to obtain reliable identification. Such challenges should likely be resolved with the generation of good quality genomes based on long-read sequencing (Chin et al., 2013).

Conclusions

We developed a pipeline that makes use of the increasing availability of whole genome sequences to identify unique taxon-discriminating genome regions as well as regions conserved across taxa that can be converted into PCR assays. We applied this approach to some of the most important plant pathogens and converted nearly one-third into qPCR assays. Our work demonstrates that a genome-wide approach can efficiently identify multiple taxon-specific genome regions that can be converted into highly specific DNA detection and identification assays.

Supplemental Information

Figure S1 Nucleotidic identity between pairwise genomes with a (A) Phytophthora dataset of 11 species and a (B) Dothideomycete dataset of 16 species

Neighbor-joining trees on the left were reconstructed from the matrix of nucleotidic identity between pairs of genomes presented on the right. For the Phytophthora dataset, the phylogenetic clades as defined in Blair et al., 2008 are indicated on the phylogenetic tree.

Click here for additional data file.

Figure S2 Relationships between the proportion of “unique” candidates and the average nucleotidic identity between genomes in the Phytophthora and Dothideomycetes datasets

(A) Unique candidates as predicted in Module 2 of the pipeline (OrthoMCL search); (B) unique candidates retained after Module 3 (filtering with BLASTp and BLASTn with e-value cutoff of 1e–05).

Click here for additional data file.

Figure S3 Proportions of “unique” candidates retained after BLASTp and BLASTn filtering according to different e-value cutoffs

(A) Filtering results for Phytophthora ramorum and P. kernoviae used as target species with the Phytophthora dataset; (B) filtering results for Sphaerulina musiva and Didymella zeae-maydis used as target species with the Dothideomycetes dataset.

Click here for additional data file.

Supplemental Information 4 Supplementary tables

Table S1. Assembly statistics and gene content for the Dothideomycete genome sequences generated or downloaded in this study. Table S2. Assembly statistics and gene content for the rust genome sequences generated or downloaded in this study. Table S3. Assembly statistics and gene content for the Phytophthora genome sequences generated or downloaded in this study.

Click here for additional data file.

Text S1 Supplementary text

Click here for additional data file.

Additional Information and Declarations

Competing Interests

Author Contributions

DNA Deposition

Data Availability

The authors declare there are no competing interests.

Nicolas Feau conceived and designed the experiments, performed the experiments, analyzed the data, prepared figures and/or tables, approved the final draft and also wrote the paper.

Stéphanie Beauseigle, Sandra Cervantes-Arango, Padmini Herath and Hesther Yueh performed the experiments, approved the final draft.

Marie-Josée Bergeron, Monique L. Sakalidis and Clement K.M. Tsui performed the experiments, analyzed the data, authored or reviewed drafts of the paper, approved the final draft.

Guillaume J. Bilodeau performed the experiments, analyzed the data, contributed reagents/materials/analysis tools, authored or reviewed drafts of the paper, approved the final draft.

Inanc Birol and Philippe Tanguay analyzed the data, contributed reagents/materials/analysis tools, approved the final draft.

Braham Dhillon and Angela L. Dale analyzed the data, authored or reviewed drafts of the paper, approved the final draft.

Steven J.M. Jones analyzed the data, contributed reagents/materials/analysis tools, authored or reviewed drafts of the paper, approved the final draft.

Josyanne Lamarche and Dario I. Ojeda performed the experiments, analyzed the data, approved the final draft.

Greg Taylor performed the experiments, analyzed the data, contributed reagents/materials/analysis tools, approved the final draft.

Adnan Uzunovic contributed reagents/materials/analysis tools, approved the final draft.

Richard C. Hamelin conceived and designed the experiments, approved the final draft and also wrote the paper.

The following information was supplied regarding the deposition of DNA sequences: Genbank Genome Assembly Accession numbers:

GCA_002116355.1, GCA_000504345.1, GCA_000504365.1, GCA_000504385.1, GCA_002116395.1, GCA_000504405.1, GCA_002116385.1, GCA_002157005.1, GCA_000204055.1, GCA_002157035.1, GCA_002157085.1, GCA_002157025.1, GCA_002157015.1, GCA_000464645.1, GCA_000500795.1, GCA_000464975.1, GCA_000500245.1, GCA_000149735.1, GCA_000500205.2, GCA_000468175.1, GCA_000448265.2.

The following information was supplied regarding data availability:

The raw data used is already public and the pipeline only uses a series of already published software or scripts; these are described in the text and the figures.

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
