# Peer review of "Genome-Enhanced Detection and Identification (GEDI) of plant pathogens"

_PeerJ, doi:10.7717/peerj.4392_

## Round 0.1 · original submission · Minor Revisions

The article is well written and addresses a relevant topic. It has been well received by three independent reviewers with only minor corrections suggested.

·

Basic reporting

The ability to correctly identify and differentiate between fungal and oomycete species is very important, with significant implications for monitoring pathogens, latent pathogens, and potential pathogens. ITS comparisons alone do not provide nearly enough information to correctly separate closely related species or species complexes, as the authors discuss. As such, use of multiple loci or equivalent screens provide a more accurate tool for differentiating species. The manuscript by Feau et al., describes a detailed pipeline that can be applied to identify taxa and/or clade specific protein-encoding sequences. It is proposed that these sequences can be used to generate qPCR markers that can be used to differentiate species. The computational analysis is supported by application of the generated qPCR primers on a wider range of isolates, supporting the claimed robustness of the assay for the species tested.

As pathogen identification and differentiation is so crucial, designing new methods is important, even if merely to open up discussions among the community about how best to advance/move away from reliance on ITS, beta-tubulin, etc. As such, and as the pipeline seems thorough and well detailed it should be considered for publication after minor revisions. However, I have some concerns and reservations that I believe should be addressed/given some thought during revisions. I outline my opinions on this matter in “general comments for the author” section below.

In general, the manuscript is clear and professional in language

Specific points:

I would remove most of the paragraph from line 73-82. Personally, I don’t think that level of detail needs to be given to outlining PCR (not even sure the acronym needs to be spelt out). Perhaps move the line “DNA detection methods have provided…. Monitoring of invasive species” down to the next paragraph, and start the next sentence along the lines of “However, …”.

Line 145: remove comma after e.g. --- (e.g., same genus…)  (e.g. same genus…)

Line 152 to 157: Confusing/convoluted sentence. I got a little lost with linking the ideas after that semi colon (line 155). Perhaps break these into two sentences, something like “de novo genome assemblies … such as ABySS (etc…)(etc…). Programs such as Augustus can provide a fast way … Gene/protein model predictions from these de novo assemblies.

--- > Further: Perhaps consider re-ordering the module outline. Almost seems that the back paragraphs are the introduction paragraphs. Or perhaps shift a few of the sentences to the front to. E.g. Next-generation sequencing (NGS) techniques constitutes…. In order to maximize the accurate….
This is stylistic, and can be ignored if the authors do not agree.

Line 172: "with a similarity cutoff defined by a e-value" --> " with a similarity cutoff defined by an e-value"

Line 194: add a comma. “unique, i.e. specific to the target…”

Line 198: add a “to”. “annotation error in target genome or due to gene…”


It is nicely spelt out in line 240 -241 “we obtained the genomes of eight non-target taxa …”. Perhaps similarly spell that out also for the dothideomycetes and rusts. E.g. obtained 39 non-target dothideomycetes.

Line 275: “five ng of DNA” change to “5 ng of DNA” in order to be consistent with the text below e.g. “concentration was at 1 uM”.

Line 287: Filtering simulations: I feel that this section lacks a little clarity. Although these results are listed in detail in the supplementary text, I think the key points summarised here could be more informative than it currently is. For example, it is slightly confusing in the first sentence starts “BLASTn and BLASTp filtering…” but then only BLASTp was mentioned later in the sentence. Since BLASTn was mentioned at the start of the sentence, it should be raised again. If you aren’t interested in discussing BLASTn results, split the sentence in two. E.g. Simulations with BLASTn and BLASTp filtering were conducted on a set of 13 Phytophthora …”. “This analysis demonstrated that BLASTp e-value cuttoff…”

Line 293: “rejected the most potential false positive”  might just be me, but that sentence doesn’t sound right. Perhaps re-word?

Line 313: Perhaps be a little more verbose here and actually state which statistics without going into too much detail (for example N50, completedness, etc.)


Check spelling of Melampsora throughout. Quite a few instances of Melamspora. E.g. in the tables.

Experimental design

No comment

Validity of the findings

No comment

Additional comments

In the discussion, there is a heavy focus on the flaws of using ITS alone. Although true, the use of ITS alone is no longer the norm. As such, it is my opinion that the paragraph discussing the negatives of ITS alone should be condensed.

When I first read the abstract, I initially interpreted lines 36-38 to mean that the authors would take the two polar approaches: 1) to look at taxa specific genes 2) to look conserved genes that could be used as more universal markers. The authors explain their intentions at the end of the introduction (110-113) much more clearly. E.g. I found that “to identify genome regions specific to targeted taxa or groups of related taxa” was clearer than “to identify unique taxon-discriminating genome regions as well as regions conserved across taxa”. Perhaps alter the abstract wording to be clearer.

I have a few major reservations with the outlined approach. These might be based on my misinterpretation, and so I would appreciate a response to these concerns or clearer discussion about potential limitations.

1) If the approach was used to screen for environmental samples, and resulted in positive amplification of the target genes, can you be sure that the sample isn’t a closely related species that has not been previously not been characterized, and with no associated genomic data? At this stage you would have to amplify and sequence other genes/sequence the genome, and by doing so, would this make the whole method redundant?
2) Gene loss is quite frequent, and so not having a gene may not actually indicate that an isolate belongs to a different species. Would there be a way to account for this/design an expected species amplification threshold for a given species? Hypothetical analysis with threshold marker; we multiplexed X number of taxon-specific genes, and successfully amplified 80% of these, which is within the acceptable range observed for trial analyses on a diverse range isolates of this species. The pipeline proposed by the authors might work for the tested species, but I wonder how it would work on a species with a high level of genetic diversity/intraspecific gene variation. For example, how would the pipeline work for a related septoria species such as Zymoseptoria, which has an incredibly high level of diversity? This is a species with a lot of genomic data available, so perhaps it could be possible to trial the robustness of the pipeline and see if it holds up on such a species?

Reviewer 2 ·

Basic reporting

I found the manuscript very carefully crafted and written.
Raw data can be retrieved from NCBI and other public genome repositories cited in the manuscript. Figures and tables are helpful and done well.

Table 2: « Number of species and genus-specific protein models found after clustering with OrthoMCL and filtering with BLASTn »
Shouldn’t it be « Number of species(…) with BLASTn and BLASTp » ?

L353: polymorphism usually refers to differences within species, not between species (usually called ‘divergence’)

Experimental design

The research question is clear, methodology is sound and well described.
Having read the first sections of the manuscript, I had several concerns with the experimental design, but I found answers to my questions in Text S1.

I have only one minor comment: Orthologous gene clusters are identified using OrthoMCL (L167). I am not a specialist such orthology analyses. But is there any reference that would support the use of OrthoMCL over other orthology analysis tools? Is there any method that could be used in place of OrthoMCL? a method that could even make the filtering steps of Module 3 unnecessary?

Validity of the findings

no comment

Additional comments

no comment

Reviewer 3 ·

Basic reporting

No comment

Experimental design

No comment

Validity of the findings

No comment

Additional comments

This manuscript describes in great detail a pipeline which can be used to develop diagnostic PCR assays for a large number of fungal and oomycete plant pathogens, deriving from full genome sequences. The bio-informatics analysis appears robust and is well described. The outcome was the development of a number of PCR assays which can detect a large number of important plant pathogens. Overall the approach is logical and these types of diagnostic assays are a useful resource for field detection of disease, and for early warning of likely disease incidence. I have only a couple of significant criticisms / questions that the authors may need to address.
1. I could see no evidence whatsoever for the functionality of the PCR assays. No gel pictures are included? I think this is necessary to support the stated conclusions, at least as supplementary material.
2. The authors have done a good job in identifying diagnostic regions of genome sequences which can distinguish different fungal species. But I could not determine whether the authors checked that these regions were conserved in the genomes of additional strains of any given species. This is important for them to be truly useful in the face the large genetic diversity (snps etc) which can be detected between different individuals of the same species. Could the authors comment on whether this was done? Or whether they believe this to be important? Perhaps some gels showing the amplicon obtained from different strains of a given pathogen would be useful?

---

## Round 0.2 · accepted · Accept

Hi Richard - I am satisfied with your responses to the reviewers and therefore happy to accept the manuscript for publication. Well done! - Peter